# Multi-Branch Network for Color Image Denoising Using Dilated Convolution and Attention Mechanisms

**DOI:** 10.3390/s24113608

**Published:** 2024-06-03

**Authors:** Minh-Thien Duong, Bao-Tran Nguyen Thi, Seongsoo Lee, Min-Cheol Hong

**Affiliations:** 1Department of Information and Telecommunication Engineering, Soongsil University, Seoul 06978, Republic of Korea; duongthien2206@soongsil.ac.kr (M.-T.D.); baotrannguyen@soongsil.ac.kr (B.-T.N.T.); 2Department of Intelligent Semiconductor, Soongsil University, Seoul 06978, Republic of Korea; sslee@ssu.ac.kr; 3School of Electronic Engineering, Soongsil University, Seoul 06978, Republic of Korea

**Keywords:** additive noise, attention mechanism, dilated convolution, multi-branch network, image denoising

## Abstract

Image denoising is regarded as an ill-posed problem in computer vision tasks that removes additive noise from imaging sensors. Recently, several convolution neural network-based image-denoising methods have achieved remarkable advances. However, it is difficult for a simple denoising network to recover aesthetically pleasing images owing to the complexity of image content. Therefore, this study proposes a multi-branch network to improve the performance of the denoising method. First, the proposed network is designed based on a conventional autoencoder to learn multi-level contextual features from input images. Subsequently, we integrate two modules into the network, including the Pyramid Context Module (PCM) and the Residual Bottleneck Attention Module (RBAM), to extract salient information for the training process. More specifically, PCM is applied at the beginning of the network to enlarge the receptive field and successfully address the loss of global information using dilated convolution. Meanwhile, RBAM is inserted into the middle of the encoder and decoder to eliminate degraded features and reduce undesired artifacts. Finally, extensive experimental results prove the superiority of the proposed method over state-of-the-art deep-learning methods in terms of objective and subjective performances.

## 1. Introduction

The image-denoising task involves reconstructing a clean image from a noisy image. Additive noise can occur for many reasons, such as camera sensors and poor lighting conditions [1,2]. It can adversely affect the quality of the captured images and the performance of vision-based intelligent systems [3,4]. Currently, owing to the tremendous success of deep-learning-based approaches in image recognition and classification tasks [5], many scientists have begun developing deep-learning-based methods for image denoising [6,7]. While it is undeniable that these methods have achieved satisfactory performance, several challenges remain that are worth studying.

First, it is difficult for a simple network to achieve high-quality image denoising because of the complexity of image content [6,7,8,9,10]. Most of these models conduct alternate downsampling and upsampling of deep features to achieve large receptive fields. Unfortunately, these alternate operations may result in significant information loss during the training process and failure to suppress noise in the recovery results. Most representatively, Solovyeva et al. [9] implement dual autoencoders for image denoising, where the first one acts as the primary image denoiser and the second ameliorates the quality of the denoised images. Nevertheless, it lacks the adaptability to resolve a given noise level, and its performance is not sufficiently impressive owing to its simple architecture. Second, global information is lost when long-term dependency structures are adopted [11,12]. Specifically, Denoising Convolutional Neural Networks (DnCNN) [11] proposes a straightforward model for image denoising. However, it sacrifices the capacity of the model and the receptive field expansion is limited. Moreover, many attempts have been made to design a deeper neural network [13] or use a non-local module to enlarge the receptive fields [14,15]; however, these methods result in high computational costs and time-consumption problems. Finally, many convolution neural network-based image-denoising methods (CNN-based methods) are subject to undesired artifacts [16,17], and some critical details are lost because they cannot adapt to textures and edges. For example, Zhang et al. [17] present a fast and flexible CNN-based method (FFDNet) for image denoising. However, the recovered images suffer from over-blurring problems owing to degraded features that are not thoroughly resolved.

As described above, deep-learning-based image-denoising methods still cause some problems: (1) it is challenging to improve the performance of the denoising task using the simple network; (2) global information is disregarded; and (3) undesired artifacts from failing to comprehensively handle degraded features.

To address these problems, we propose a multi-branch network based on a conventional autoencoder to learn multi-level contextual features from noisy images. Moreover, we integrate two blocks, including the Pyramid Context Module (PCM) and the Residual Bottleneck Attention Module (RBAM), into the network to select salient information during the training process. The architecture of the proposed network is illustrated in Figure 1. More specifically, a multi-branch network has a structure that combines multi-level contextual feature maps using a skip connection. It has been proven that this structure is efficient for learning via structural analysis of the image and robustly eliminates additive noise in the image [18]. Additionally, a PCM is added at the beginning of the network to handle the global information loss problem. It uses a parallel-dilated convolution operation with four dilation rates and is arranged in a pyramid form. Last but not least, we insert the RBAM into the middle of the encoder and decoder to focus on valuable features and neglect degraded features without introducing excessive additional computation. The performance of the proposed method is evaluated using various quantitative metrics.

To sum up, the main contributions of this paper are listed as follows:A multi-branch network that effectively improves the performance of image-denoising tasks is presented.A PCM that uses dilated convolution is introduced to enlarge the receptive field and successfully address the loss of global information.An RBAM is designed to eliminate degraded features and reduce undesired artifacts.Comprehensive experiments are performed on several datasets, proving that the proposed method surpasses other competitive methods.

The remainder of this paper is organized as follows: Section 2 provides a brief review of related works. In Section 3, the proposed method is introduced. Section 4 reports the experimental results of the proposed method, and Section 5 presents the conclusions.

## 2. Related Work

Image denoising aims to recover a clean image *x* from a noisy image *y*. Generally, the degradation model is formulated as *y* = *x* + *n* where *n* indicates the additive noise. Image-denoising methods can be divided into two major categories: traditional and deep-learning methods.

Traditional methods can flexibly solve denoising problems with different noise levels. For example, BM3D [19] is a mainstream method that enhances sparsity by grouping similar 2-D image fragments into 3-D data arrays. Chang et al. [20] present an adaptive data-driven threshold to decouple the noisy image into frequency bands and apply a threshold to suppress noise. The Wiener filter has been applied for the removal of Gaussian noise to address the drawback of the mean filter, which is susceptible to image over-smoothing with a high noise level [21,22]. Median, weighted-median, and bilateral filters can minimize additive noise without special identification because of their edge-preserving properties [23,24,25]. The total variation is based on the integral of the absolute image gradient, which increases when the images contain immense detail; in particular, it decreases noise while preserving the borders of the image [26,27]. Overall, the performance of these methods depends on their optimization algorithms, which should carefully select the parameters, and the computational cost is significantly high.

Recently, deep-learning methods have successfully handled image denoising [6,7]. One of the earliest attempts, DnCNN [11], proposes residual learning and batch normalization to implement end-to-end image denoising. Regarding prior CNN-based denoiser approaches, DRUNet [12] is a reliable CNN-based option that has shown great promise for ill-posed problems and for developing a powerful, adaptable solution. Moreover, a deep CNN denoiser prior for image restoration (IRCNN) [16] uses known noise levels to train a denoiser and then leverages this denoiser to estimate the noise level. To improve denoising speed, FFDNet [17] utilizes noise levels and noisy images as inputs for a CNN-based network. RDUNet [28] is a residual dense neural network for image denoising based on a densely connected hierarchical network. Recently, transformer technology has been applied to image denoising [29,30]. Most representatively, swin-transformer UNet for image denoising (SUNet) [31] and swin-transformer-based image restoration (SwinIR) [32] adopt the swin-transformer as the primary module and integrate it into a unique denoising architecture to suppress additive noise. Furthermore, Xia et al. [33] introduce an efficient diffusion model for image restoration (DiffIR), which contains a compact IR prior extraction network (CPEN), dynamic IR transformer (DIRformer), and denoising network. Yang et al. [34] propose an approach for real-world denoising based on a general diffusion model with linear interpolation. MambaIR [35] improves the vanilla Mamba model using a Residual State Space block with both local convolution-based enhancement and channel attention for image-denoising tasks.

In short, deep learning methods outperform traditional methods to a certain extent. However, there are many methods to improve denoising performance, particularly by focusing on global information and reducing undesired artifacts using an efficient deep-learning network. To this end, we devised a multi-branch network using dilated convolution and an attention mechanism that enriches global information and eliminates degraded features.

## 3. Proposed Method

In this section, we present the architecture of a multi-branch network in conjunction with two feature extraction modules, including the Pyramid Context Module (PCM) and the Residual Bottleneck Attention Module (RBAM). Subsequently, the loss function is also introduced to optimize the proposed network.

### 3.1. Network Architecture

Figure 1 depicts the structure of the proposed network. First, a multi-branch network based on a conventional autoencoder architecture for image denoising is proposed. Second, we added a PCM to the beginning of the network to extract useful global information. Ultimately, the RBAM is inserted into the middle of the encoder and decoder to filter undesirable artifacts. The convolutional layers at the start and end assist the network in capturing complicated mappings between the image and its features.

#### 3.1.1. Multi-Branch Network

Many scientists recently employed autoencoder structures based on convolutional neural networks (CNNs) to minimize additive noise [6,7,36]. Most of these methods adopt an encoder–decoder framework to learn the features of various receptive fields. Nevertheless, the repeated upsampling and downsampling operations contained in the encoder–decoder framework encounter a loss of texture details, seriously affecting the restoration of the image. To address this problem, we devise a multi-branch network based on a conventional autoencoder architecture for image denoising. This network combines multi-level contextual feature maps using skip connections. It has been proven that the structure can be easily learned via structural analysis of the image and effectively removes additive noise from the image [18]. The proposed network has three scales in each encoder–decoder convolutional module. We utilize downsampling with a stride convolutional layer in the encoder to compress essential information. In the decoder, we apply a resize-convolutional layer [37] for upsampling and achieve a feature map size commensurate with its mirror in the encoder part. Skip connections are used between the encoder and the corresponding decoder blocks aiming to reconstruct features and image information that are typically lost during the encoding stage. However, it is noteworthy that it is only used for the first branch. In addition, because the low-level features extracted before a noisy image is fed into the multi-branch network contain immense color information, we concatenate them with the last feature map of the three branches via a global skip connection. Subsequently, a 3 × 3 convolutional layer is used to fuse the previously extracted low-level features and noise-free high-level features to generate the output image. The sigmoid function is used in the proposed network to introduce the non-linearity property and its output in the range 0 to 1. In addition, we adopt a normalization operator to rescale the input images between 0 and 1 before training the network. This helps to stabilize the gradient descent step, allowing the network to use larger learning rates and converge faster for a given learning rate. After the training process is completed, the output images will be rescaled back to their original pixel values to produce color images. These modifications promise to improve the performance of our network in promoting image denoising and preventing information loss during the image restoration process.

#### 3.1.2. Pyramid Context Module (PCM)

As mentioned previously, global information is typically lost in an autoencoder-based model because the receptive field expansion is limited. Inspired by [38], dilated convolution is a filter expansion technique used in convolutional neural networks (CNNs). In this technique, the filter has gaps between its elements, determined by a dilation rate (DR). Dilated convolution helps increase the receptive field of the network without significantly increasing parameters, allowing the network to capture more global information from the input data. It is useful in tasks where capturing contextual information over a large spatial extent is important, particularly image restoration. Therefore, we introduced the Pyramid Context Module (PCM) using dilated convolution and inserted it at the beginning of the network to obtain abundant receptive field information, as shown in Figure 2. Specifically, we employ parallel dilated convolutions to extract multi-context features inspired by the Atrous Spatial Pyramid Pooling (ASPP) block [39]. This enables the networks to learn context-sensitive information. The feature pyramid is enriched by concatenating the outputs of all the parallel dilated convolutional layers. These parallel layers have progressively wider contexts because of the rising dilation rates (DR = 1, 2, 3, and 4). We then adopt a 1 × 1 convolutional layer to fuse features from various receptive fields. Additionally, we apply long-skip connections to leverage information from shallow features. Finally, the fused features are combined with the input features to obtain the output using an element-wise addition operation.

#### 3.1.3. Residual Bottleneck Attention Module (RBAM)

Although the autoencoder structure with symmetric skip connections has shown promising performance for image-denoising tasks [9], we observed that some undesired artifacts remained in the final results. One justifiable reason for this problem is that the degraded features are passed from the encoder to the decoder. To overcome this problem, we leverage the attention mechanism [40], which is widely used for diverse image restoration tasks [41,42], into the middle of the encoder and decoder. In this study, we present an RBAM composed of two branches, the Channel Attention Module (CAM) and the Spatial Attention Module (SAM), to eliminate degraded features with the aim of reducing undesired artifacts, as depicted in Figure 3. This module was inspired by [43] and proved without introducing excessive computation. However, unlike the previous method, one of the most significant changes in our RBAM is residual learning [44], which prevents the vanishing gradient problem and is robust for training processing.

For the input feature map $F\in R^{H_{i}\times W_{i}\times C_{i}}$, RBAM produces an attention map $M\left(F\right)\in R^{H_{i}\times W_{i}\times C_{i}}$. Here, $H_{i},W_{i},$ and $C_{i}$ denotes the height, width, and channels of the *i*th feature map, respectively. The output feature map $F^{\prime }\in R^{H_{i}\times W_{i}\times C_{i}}$ is expressed as
(1)$$F^{\prime }=conv_{3\times 3}\left(conv_{3\times 3}\left(conv_{3\times 3}\left(F\right)\right)\right)+F\circ M\left(F\right),$$
where ∘ symbolizes the element-wise product.

We utilize residual learning in conjunction with attention mechanisms to improve gradient flow. To design an efficient yet robust module, we first calculate the channel attention $M_{c}\left(F\right)\in R^{1\times 1\times C_{i}}$ and spatial attention $M_{s}\left(F\right)\in R^{H_{i}\times W_{i}\times 1}$ at two segregated branches. Finally, the attention map M(F) is computed as
(2)$$M\left(F\right)=\sigma (M_{c}\left(F\right)+M_{s}\left(F\right)),$$
where σ represents a sigmoid function.

In RBAM, the CAM produces a channel attention map $M_{c}\left(F\right)$ to focus on the meaning of the input features, while SAM generates a spatial attention map $M_{s}\left(F\right)$ that concentrates on the position of the informative part. In particular, the channel and spatial attention maps are calculated as follows:(3)$$M_{c}\left(F\right)=BN\left(MLP\left(GAP\left(F\right)\right)\right),$$
(4)$$M_{s}\left(F\right)=BN\left(conv_{1\times 1}\left(conv_{3\times 3}\left(conv_{3\times 3}\left(conv_{1\times 1}\left(F\right)\right)\right)\right)\right),$$
where BN(·) is the batch normalization layer [45], and MLP(·) represents a multi-layer perceptron with one hidden layer. GAP(·) denotes the global average pooling on the feature map *F* to generate a channel vector.

### 3.2. Loss Function

In this section, a three-term loss function is proposed that consists of Charbonnier loss [46], structure loss, and perceptual loss [47]. The total loss function can be represented as
(5)$$L_{total}=L_{char}+L_{str}+L_{per}.$$

The Charbonnier loss function is used to measure the difference between the denoised image and the ground-truth image. Compared to the $L_{1}$ loss function, it may better settle outliers and enhance model performance. It is defined as
(6)$$L_{char}=\sqrt{{\left\|X-Y\right\|}^{2}+\epsilon^{2}},$$
where *X* and *Y* represent the noise-free and ground-truth images, respectively. ϵ is the coefficient for which the loss function changes from approximately quadratic to approximately linear.

In addition, the structural similarity index matrix (SSIM) compares the similarities between two images. To better preserve the structural information of an image, we used the structural loss as the SSIM loss [48], which is expressed as
(7)$$L_{str}=1-SSIM\left(X,Y\right),$$
where SSIM(·) indicates the SSIM operator.
(8)$$SSIM\left(x,y\right)=\frac{\left(2\;\mu_{x}\mu_{y}+C_{1}\right)\left(2\sigma_{xy}+C_{2}\right)}{\left(\mu_{x}^{2}+\mu_{y}^{2}+C_{1}\right)\left(\sigma_{x}^{2}+\sigma_{y}^{2}+C_{2}\right)},$$
where $\mu_{x}$ and $\mu_{y}$ are the mean values of two images; $\sigma_{x}$ and $\sigma_{y}$ are variance values of two images, $\sigma_{xy}$ is the covariance between the two images, and $C_{1}$ and $C_{2}$ are two constants that prevent the denominator from being zero.

Generally, per-pixel-based loss functions make it difficult to determine the differences in high-level features between noise-free and ground-truth images. This study utilizes the perceptual loss function to minimize the differences in high-level features extracted from the pre-trained VGG-16 network [49], which can be formulated as
(9)$$L_{per}=\frac{1}{W_{n}H_{n}C_{n}}{\left(\|\rho \left(X\right)-\rho \left(Y\right)\right\|}_{2}^{2}),$$
where ρ(·) represents the feature map obtained from the VGG-16 network. $W_{n}$, $H_{n}$, $C_{n}$ denote the width, height, and the number of channels of the corresponding feature maps, respectively.

## 4. Experimental Results

In this section, we first present the experimental settings, including the datasets, training information, and experimental environment. Subsequently, the experimental results are analyzed and discussed to demonstrate the capability of the proposed network.

### 4.1. Experimental Setting

Following the previous studies [16,28,31], we generate a synthetic noisy image by adding additive white Gaussian noise (AWGN) with a wide range noise level σ∈[5,50] for 800 images from the DIV2K [50] with average spatial resolution of 1920 × 1080 for the training process. In addition, we trained the proposed network with 320 real noisy images from SIDD [51]. The evaluation process is examined on 100 images with characteristics similar to those of the DIV2K training dataset. The training process is optimized by minimizing the total loss function for 200 epochs using the Adam optimizer [52]. The coefficient ϵ in Equation (Equation 6) was empirically set to 0.001. The patch size and batch size are set to 64 × 64 and 24, respectively. All experiments were performed using the PyTorch 1.12.1 library on an NVIDIA RTX 3090 GPU.

For the testing datasets, we utilized CBSD68 [53], which contains 68 images with image size 768 × 512, and the Kodak24 dataset [54], containing 24 images with the spatial resolution of 321 × 481. Additionally, we also evaluated the proposed network on two real noisy datasets: SIDD [51] and RNI15 [55]. It is worth noting that RNI15 encompasses different types of real noise, including camera noise and JPEG compression noise. Because ground-truth images are unavailable, we can only illustrate subjective comparisons on these images. The proposed method was compared with several state-of-the-art deep-learning methods such as DnCNN [11], DRUNet [12], IRCNN [16], FFDNet [17], RDUNet [28], SUNet [31], and SwinIR [32].

### 4.2. Analysis of Experimental Results

To evaluate the performance of the proposed method, we employed the peak signal-to-noise ratio (PSNR) and structural similarity index (SSIM) metrics. The two evaluators calculated the difference between the clean and ground-truth images in a pixel-wise manner, and the highest score indicated the best performance. For comparison purposes, we report the results for noise levels σ={15,25,50}, which are the most widely used in the literature [16,32]. The PSNR/SSIM results of the different methods on the DIV2K [50], CBSD68 [53], and Kodak24 [54] datasets are listed in Table 1 and Table 2. The results show that the proposed method achieves better performance than all competitive methods. In particular, the PSNR improved by 0.11∼1.09 dB (decibels), 0.13∼1.48 dB, 0.18∼2.44 dB on the DIV2K validation dataset, 0.05∼1.22 dB, 0.27∼0.92 dB, 0.07∼0.78 dB on the CBSD68 dataset, and 0.01∼1.65 dB, 0.19∼1.97 dB, 0.01∼0.94 dB on the Kodak24 dataset for noisy level σ={15,25,50} compared to the competitive methods, respectively. In addition, the proposed method achieves the highest performance in terms of the SSIM metric against competitive methods with noise levels of 25 and 50, except for a noise level of 15 at the second rank. For example, the SSIM improved by 0.021∼0.048 on the DIV2K, 0.001∼0.032 on the CBSD68, and 0.002∼0.042 on the Kodak24 datasets compared to competitive methods with a noise level of 50, which demonstrates that the proposed method efficiently eliminates the heavy noise level.

Table 3 compares the complexity of the different methods in terms of the number of trainable parameters. Evidently, methods such as DnCNN, FFDNet, IRCNN, and SwinIR have smaller parameters because of the single network architecture, and SwinIR does not perform any downsampling operation. Therefore, it is difficult to recover high-quality images owing to the complexity of the image content. By contrast, SUNet, RDUNet, and DRUNet have larger parameters because they apply a transformer technique or many up/down-sampling operations. Although our proposed method uses a multi-branch network, the number of parameters is acceptable. The proposed method only has 21 M parameters, whereas RDUNet has 166 M parameters, but our network surpasses RDUNet by up to 0.50 dB on the DIV2K validation dataset, 0.26 dB on the CBSD68 dataset, and 0.29 dB on the Kodak24 dataset for noise level 50. This indicates that the proposed method is significantly efficient for image denoising with moderate model size.

Furthermore, Table 4 indicates the running time of different methods on images of size 256 × 256 with noise level 25. It is worth noting that the running time depends significantly on the hardware, especially GPU computations. The running time comparison was implemented on an NVIDIA RTX 3090 GPU. According to the results, the running time of FFDNet is the fastest since it works on downsampled sub-images, followed closely by IRCNN and DnCNN methods. Due to the computational complexity, DRUNet required longer running times than the compared methods. Meanwhile, the proposed method obtains a good trade-off between inference speed and denoising performance. In summary, this confirms that the proposed method has great potential to meet the practical requirements.

The visual comparison results for color image denoising by the different methods are depicted in Figure 4, Figure 5, Figure 6 and Figure 7. Furthermore, we partially zoomed in to view a certain region from a clean image to clarify the visual comparisons with competitive methods. These figures indicate that our architecture is highly efficient in learning feature representations in image-denoising tasks. As shown in Figure 4, several methods such as DnCNN, DRUNet, IRCNN, and SUNet fail to remove noise and recover rich textures. The results of FFDNet, SwinIR, and RDUNet are prone to over-smoothing problems and loss of detailed structures. On the contrary, the proposed method eliminates additive noise, preserves image details, and yields sharper edges. Looking more closely at Figure 5, it can be seen that the DnCNN, SUNet, SwinIR, and IRCNN methods are susceptible to image distortions and unwanted artifacts. FFDNet and DRUNet fail to restore aesthetically pleasing images, whereas RDUNet leaves noise in the final image. In contrast, our proposed method better preserves the texture and structural patterns of clean images. Figure 6 illustrates the visual results for an image from the SIDD real noisy dataset. The DnCNN, IRCNN, FFDNet, and SUNet suffer from confusing artifacts with splotchy textures. Additionally, SwinIR and RDUNet contain unfavorable effects and image distortions, while the DRUNet methods lead to over-smoothing of the contents. By contrast, the proposed network can better preserve fine textures and structures that are closer to the ground-truth. In addition, we also present denoising results on the RNI15 real noisy image in Figure 7. These figures indicate that our method can suppress the noise without introducing any artifacts. Based on the experimental results, noise removal and unexpected artifact reduction should be considered jointly in image-denoising tasks. The experiments reveal that the multi-branch network using dilated convolution and the attention mechanism obtained subjectively and objectively exhibits outstanding performance.

### 4.3. Ablation Study

In this section, we carry out some experiments to emphasize the effectiveness of different modules in the proposed network. Table 5 illustrates a performance comparison of the ablation study based on the different configurations ($V_{1}$ and $V_{2}$) on the DIV2K validation dataset at noise level 50.

#### 4.3.1. Effectiveness of PCM

We evaluate the usefulness of the PCM by omitting the PCM (defined as w/o PCM) in the proposed network. The result indicates that the network with PCM outperforms that without PCM by 0.0172 in terms of SSIM and 1.30 dB in terms of PSNR. In summary, the ablation result demonstrates that the PCM can efficiently capture global and contextual information.

#### 4.3.2. Effectiveness of RBAM

We assess the superiority of the RBAM and compared it with a single convolutional layer (defined as SCL). We then substitute the RBAM with the SCL in the network. Our proposed RBAM surpasses the SCL by 0.0215 regarding SSIM and 0.91 dB regarding PSNR. Accordingly, the ablation result implies that the proposed RBAM can efficiently extract salient features and ignore degraded features.

## 5. Conclusions

In this study, a multi-branch network for color image denoising using dilated convolution and attention modules is presented. The proposed method enriches global information and eliminates degraded features during the image-denoising process. The experiments demonstrated that the proposed method obtains subjectively and objectively promising results compared with other state-of-the-art deep-learning methods. In particular, it was verified that the proposed method could effectively suppress additive noise and reduce undesired artifacts. Currently, the method under investigation applies other image restoration tasks simultaneously to obtain high-quality images and is expected to achieve promising performance.

## Figures and Tables

**Figure 1 sensors-24-03608-f001:**
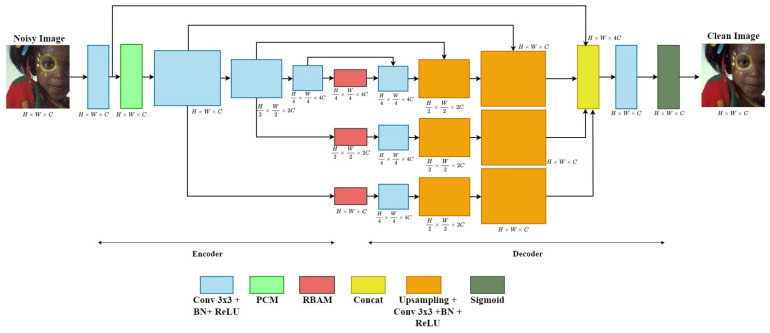
Architecture of proposed network.

**Figure 2 sensors-24-03608-f002:**
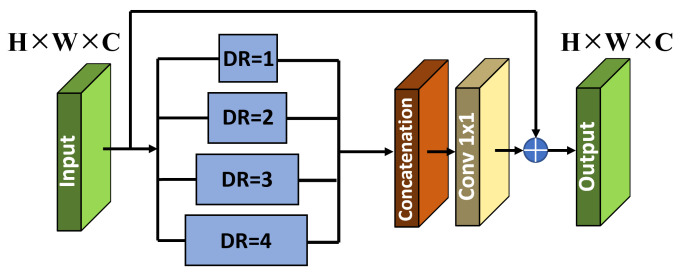
Architecture of Pyramid Context Module (PCM).

**Figure 3 sensors-24-03608-f003:**
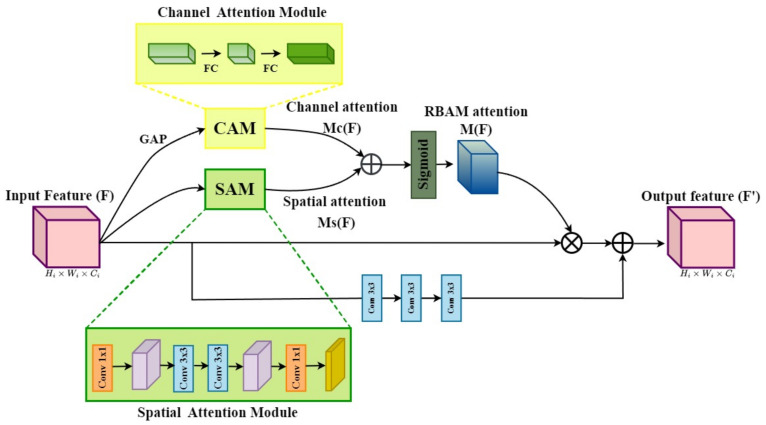
Architecture of Residual Bottleneck Attention Module (RBAM).

**Figure 4 sensors-24-03608-f004:**
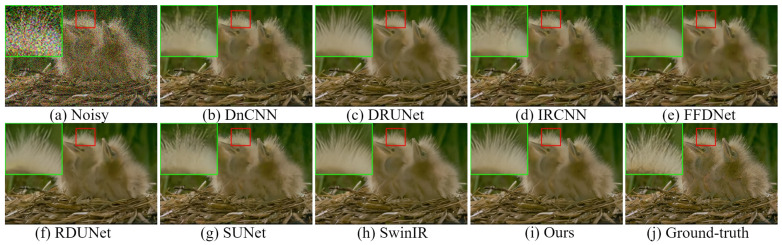
Visual comparisons of denoising results on a representative image from the CBSD68 dataset with noise level 50.

**Figure 5 sensors-24-03608-f005:**
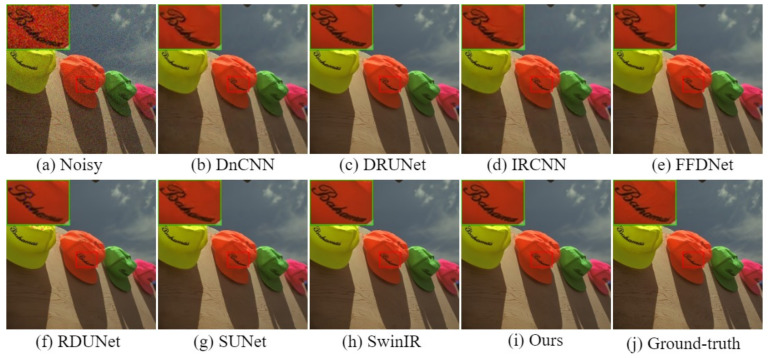
Visual comparisons of denoising results on a representative image from the Kodak24 dataset with noise level 25.

**Figure 6 sensors-24-03608-f006:**
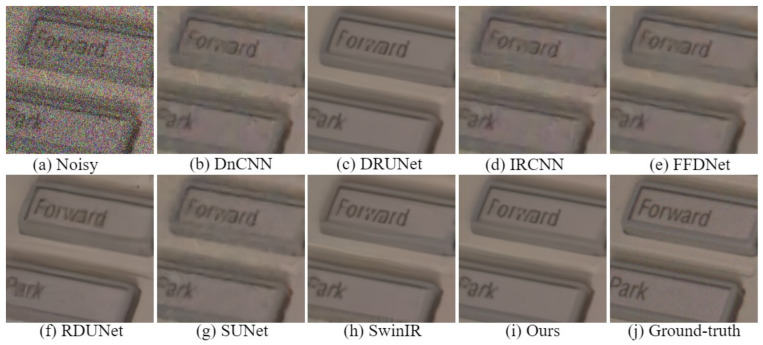
Visual comparisons of denoising results on a representative image from the SIDD real noisy dataset.

**Figure 7 sensors-24-03608-f007:**
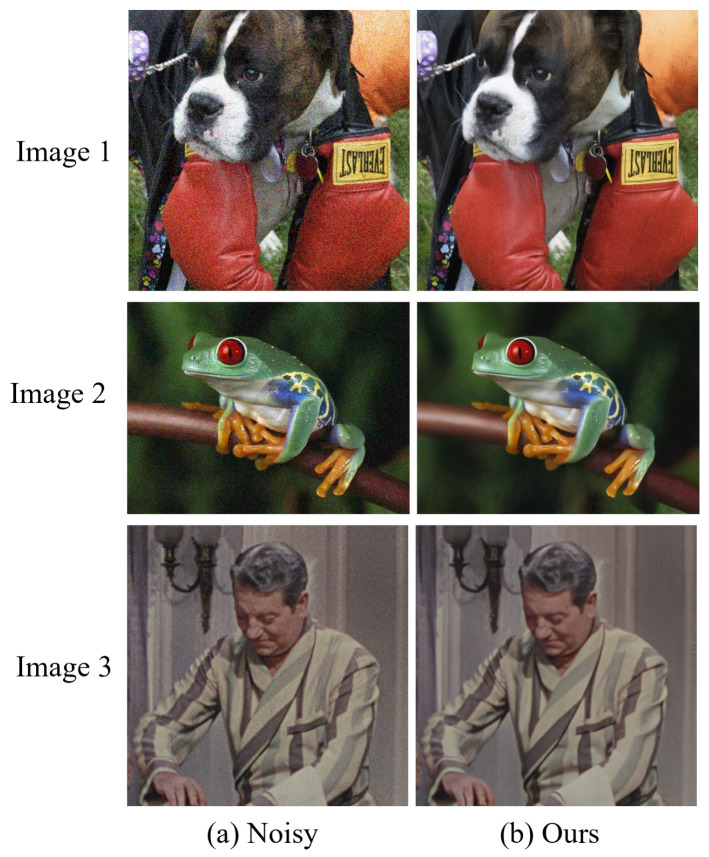
Denoising results of proposed network on representative images from the RNI15 real noisy dataset.

**Table 1 sensors-24-03608-t001:** Average PSNR (dB) results of different methods for color image denoising with noise levels 15, 25, and 50 on the DIV2K, CBSD68, and Kodak24 datasets. The best and second-best results are highlighted in **red** and **blue** colors, respectively.

Methods	DIV2K Validation Dataset [50]			CBSD68 Dataset [53]			Kodak24 Dataset [54]		
	σ=15	σ=25	σ=50	σ=15	σ=25	σ=50	σ=15	σ=25	σ=50
DnCNN [11]	39.86	38.35	35.63	33.90	31.24	27.95	34.60	32.14	28.95
DRUNet [12]	**40.79**	**39.63**	**37.89**	34.30	31.69	28.51	35.31	**32.89**	**29.86**
IRCNN [16]	39.81	38.28	35.65	33.86	31.16	27.86	34.69	32.18	28.93
FFDNet [17]	40.18	38.79	36.45	33.87	31.21	27.96	34.63	32.13	28.98
RDUNet [28]	40.69	39.48	37.57	34.24	31.60	28.37	35.13	32.69	29.58
SUNet [31]	40.30	39.08	36.96	33.25	31.13	27.85	33.67	31.11	29.54
SwinIR [32]	**40.79**	39.58	37.78	**34.42**	**31.78**	**28.56**	**35.34**	**32.89**	29.79
Ours	**40.90**	**39.76**	**38.07**	**34.47**	**32.05**	**28.63**	**35.32**	**33.08**	**29.87**

**Table 2 sensors-24-03608-t002:** Average SSIM results of different methods for color image denoising with noise levels 15, 25, and 50 on the DIV2K, CBSD68, and Kodak24 datasets. The best and second-best results are highlighted in **red** and **blue** colors, respectively.

Methods	DIV2K Validation Dataset [50]			CBSD68 Dataset [53]			Kodak24 Dataset [54]		
	σ=15	σ=25	σ=50	σ=15	σ=25	σ=50	σ=15	σ=25	σ=50
DnCNN [11]	0.9246	0.9095	0.8797	0.9290	0.8830	0.7896	0.8763	0.8823	0.7808
DRUNet [12]	0.9345	0.9209	0.9037	0.9344	0.8926	**0.8199**	0.9287	0.8912	0.8199
IRCNN [16]	0.9243	0.9086	0.8809	0.9285	0.8824	0.7898	0.9198	0.8766	0.7929
FFDNet [17]	0.9272	0.9129	0.8901	0.9290	0.8821	0.7887	0.9215	0.8779	0.7942
RDUNet [28]	0.9330	0.9193	0.9007	0.9340	0.8912	0.8062	0.9287	0.8903	0.8171
SUNet [31]	**0.9536**	**0.9225**	**0.9059**	**0.9372**	0.8869	0.7995	**0.9308**	**0.9014**	0.8105
SwinIR [32]	0.9351	0.9213	0.9033	0.9350	**0.8940**	0.8119	0.9300	0.8927	**0.8216**
Ours	**0.9357**	**0.9426**	**0.9277**	**0.9356**	**0.8942**	**0.8210**	**0.9304**	**0.9189**	**0.8232**

**Table 3 sensors-24-03608-t003:** Objective comparisons of computational complexity in terms of the number of trainable parameters.

Methods	DnCNN [11]	DRUNet [12]	IRCNN [16]	FFDNet [17]	RDUNet [28]	SUNet [31]	SwinIR [32]	Ours
Parameters	558 K	32.64 M	420 K	854 K	166 M	99 M	12 M	21 M

**Table 4 sensors-24-03608-t004:** Running time (in seconds) of different methods on images of size 256 × 256 with noise level 25.

Methods	DnCNN [11]	DRUNet [12]	IRCNN [16]	FFDNet [17]	Ours
Running time	0.0087	0.0221	0.0066	0.0023	0.0094

**Table 5 sensors-24-03608-t005:** Ablation experiment results of different configurations on DIV2K validation dataset at noise level 50. The symbol “↑” represents the higher, the better.

Defined	Configuration	SSIM ↑	PSNR ↑
$V_{1}$	PCM → w/o PCM	0.9105	36.77
$V_{2}$	RBAM → SCL	0.9062	37.16
**Ours**	Default	**0.9277**	**38.07**

## Data Availability

Data are contained within the article.

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
