# Peer review of "Multi-Branch Network for Color Image Denoising Using Dilated Convolution and Attention Mechanisms"

_sensors, 2024, doi:10.3390/s24113608_

Round 1

Reviewer 1 Report

Comments and Suggestions for Authors

1. The paper needs to further clarify why existing solutions are insufficient to address the target problem, as well as theoretically elaborate on the main advantages of the new network structure.

2. Please provide a more detailed description of the network architecture, including details of inputs and outputs, the types of each layer, etc. A deeper explanation is required for key or innovative layers proposed in the manuscript. Besides, key modules are only represented as blank blocks in Fig. 1; specific details should be provided.

3. The paper should also introduce information about network training, which is crucial for replicating the model and understanding its performance. This includes the datasets used, training parameters (such as hyperparameters), training equipment, and training results. To enhance the transparency and reliability of the research, it is recommended to provide access to training code.

4. If the author believes that the Pyramid Context Module (PCM) and the Residual Bottleneck Attention Module are key factors in enhancing model performance, it is advisable to conduct ablation experiments. These experiments can validate the specific contributions of each component and their impact on final performance, thereby confirming the effectiveness of these modules.

Comments on the Quality of English Language

The paper is well-organized and cohesive.

Reviewer 2 Report

Comments and Suggestions for Authors

This paper presents a multi-branch network for color denoising using dilated convolution and attention modules. The network learns multi-level contextual features from input images and extract salient information for training. The proposed method enhances global information and eliminates degraded features, outperforming other deep-learning methods. Experiments show the method effectively suppresses additive noise and reduces unwanted artifacts. The method is currently investigated for simultaneous image restoration tasks, aiming to promising performance. The paper is well structured and well organized and most references are included. Some comments arise after reviewing the paper:

  1. Please add an explanation of “Dilated Convolution”.
  2. Since the attention mechanism is an important component of the modal, please add more details and explanation how it was used.
  3. Include the PCM and RBAM references.
  4. In Fig 1: architecture of proposed method, please add missing text for boxes, like after PCM and RBAM blue boxes.
  5. In the same Fig 1 specify the encoder and decoder part.
  6. Fig 1 shows that the sigmoid function is used as activation function in the last layer, and the output  is in range of 0 to 1, please describe how the final image is reconstructed.
  7. Add a summary architecture that shows shapes and number of training parameters.
  8. Add input and output shape to Fig 2 and Fig 3.
  9. Add missing keys to Fig 3.
  10. Please verify formula 4. Maybe the number 2 is a square number.
  11. Add SSIM formula after formula 3.
  12. Explain what dB means.

Reviewer 3 Report

Comments and Suggestions for Authors

In the paper, the authors study the problem of image denoising. They propose a new method in the class of methods based on deep learning. In the proposed method, the authors use a multi-branch network in which dilated convolution and attention modules are introduced. The proposed method is compared with other state-of-the-art methods in the deep learning class of methods. The experiments show that the method proposed by the authors is better than the other methods. In general, the paper is interesting and well-written. There are only some minor issues that I want to raise:
1. Some of the abbreviations are not expanded at their first occurrence, e.g., DnCNN, CNN, DDFNet etc. Each abbreviation should have it extended form at the first occurrence, so the authors should go through the paper and correct this.
2. In Fig. 1, some of the blocks are empty. Why are they empty? What should be inside them?
3. We cannot start a subsection right after starting a section. There should be at least one paragraph of text. Thus, the authors should add the missing paragraph between 3 and 3.1, and between 4 and 4.1.
4. In eq. (3) the authors use the SSIM operator, but they do not define it. They should include the definition of this operator in the paper.
5. In the experimental section, we do not find any experiments regarding the computation times, which is also a very important factor in assessing the methods. Thus, the authors should add a comparison of the times.

Reviewer 4 Report

Comments and Suggestions for Authors

This paper proposes a multi-branch network to improve the performance of image denoising task. While experimental results have shown some effectiveness, it is important to note that there are still several issues that require attention and resolution.

1.       The introduction is poorly written with weak logic. I suggest integrating the "related work" section into the introduction to logically outline the issues in image denoising task, previous methods proposed in research to address these issues, and the problems that still need to be resolved.

2.       The review of image denoising methods in the introduction and related work sections is not thorough enough. Classic methods in the field are not adequately referenced, and there is limited coverage of the latest transformer-based, diffusion model-based, and mamba model-based methods, etc.

3.       The novelty in this study appears to be somewhat limited. The proposed model appears to be a simple combination of previously proposed methods.

4.       The experimental section of this paper has significant shortcomings, particularly the lack of ablation study to validate the effectiveness of different modules in the proposed model.

5.       More tests should be conducted on real-world images to demonstrate the efficacy of the proposed method.

Comments on the Quality of English Language

Minor editing of English language required.

Round 2

Reviewer 4 Report

Comments and Suggestions for Authors

To improve the performance of color image denoising task, a multi-branch network is proposed. Most of the previous problems have been resolved, but there are still several issues that need to be resolved.

1) The logic of the abstract needs to be strengthened, and the abstract needs to be more concise.

2) The difference between the proposed model and the previously proposed method is not obvious. It is suggested to add a paragraph to analyze the differences between the proposed method and the existing method to reflect the contribution of this paper.

Comments on the Quality of English Language

Minor editing of English language required.
